

# BMI prediction within a Korean population

Jin Sol Lee[1,2], Hyun Sub Cheong[3] and Hyoung-Doo Shin[1,2,3]

[1] Research Institute for Basic Science, Sogang University, Seoul, Republic of Korea
[2] Department of Life Science, Sogang University, Seoul, South Korea
[3] Department of Genetic Epidemiology, SNP Genetics, Inc., Seoul, Republic of Korea

## ABSTRACT

**Background**. Body Mass Index (BMI) is widely regarded as an important clinical trait for obesity and other diseases such as Type 2 diabetes, coronary heart disease, and osteoarthritis.

**Methods**. This study uses 6,011 samples of genotype data from ethnic Korean subjects. The data was retrieved from the Korea Association Resource. To identify the BMI-related markers within the Korean population, we collected genome-wide association study (GWAS) markers using a GWAS catalog and also obtained other markers from nearby regions. Of the total 6,011 samples, 5,410 subjects were used as part of a single nucleotide polymorphism (SNP) selection set in order to identify the overlapping BMI-associated SNPs within a 10-fold cross validation.

**Results**. We selected nine SNPs (*rs12566985* (*FPGT-TNNI3K*), *rs6545809* (*ADCY3*), *rs2943634* (located near *LOC646736*), *rs734597* (located near *TFAP2B*), *rs11030104* (*BDNF*), *rs7988412* (*GTF3A*), *rs2241423* (*MAP2K5*), *rs7202116* (*FTO*), and *rs6567160* (located near *LOC105372152*) to assist in BMI prediction. The calculated weighted genetic risk scores based on the selected 9 SNPs within the SNP selection set were applied to the final validation set consisting of 601 samples. Our results showed upward trends in the BMI values ($P < 0.0001$) within the 10-fold cross validation process for $R^2 > 0.22$. These trends were also observed within the validation set for all subjects, as well as within the validation sets divided by gender ($P < 0.0001$, $R^2 > 0.46$).

**Discussion**. The set of nine SNPs identified in this study may be useful for prospective predictions of BMI.

Corresponding authors
Hyun Sub Cheong,
chhs@snp-genetics.com
Hyoung-Doo Shin,
hdshin@sogang.ac.kr

## INTRODUCTION

Body Mass Index (BMI) is widely used as a diagnostic measurement of obesity which in turn is related to various diseases such as heart disease, Type 2 diabetes, hypertension, and osteoarthritis *(1998)*. Because of the important role BMI plays, numerous genome-wide association studies (GWAS) have been conducted to identify BMI-associated single nucleotide polymorphisms (SNPs). From these studies, researchers have identified several significant genes found to be related to BMI such as *FTO*, *BDNF*, and *MC4R*, among others (*Felix et al., 2016*; *Locke et al., 2015*; *Speliotes et al., 2010*). Further replication studies across various populations have also reported significant associations between these genes

and BMI (*Bonaccorso et al., 2015*; *Munoz-Yanez et al., 2016*; *Neocleous et al., 2016*). Using these results, several studies have developed prediction models for BMI and obesity among various ethnic groups (*Bae et al., 2016*; *Hung et al., 2015*; *Peterson et al., 2014*).

Although several BMI or obesity prediction models have been developed, these models have not been applicable to the Korean population. One issue is the significant genetic difference between the Korean population and other populations. The significance of many BMI-associated SNPs in GWAS conducted on other populations have not been replicated in the Korean population. Moreover, although SNPs in the prediction models have shown significant associations with BMI, the effective size difference of these prediction models are unsuitable for the Korean population. These results suggest a need for new BMI prediction methods that can be applied to the Korean population.

There are several methods for predicting specific diseases or clinical traits using multiple loci. The weighted genetic risk score (wGRS) is one simple and effective method for constructing SNP sets integrating the multiplied risk allele number of each SNP with the regression coefficient. Several previous studies have shown the effectiveness of using the wGRS to build a prediction model for various diseases (*Chen et al., 2011*; *Palmieri et al., 2017*; *Thanassoulis et al., 2012*).

In the present study, we aimed to identify BMI-associated SNPs for the Korean population using genotype data obtained from the Korea Association Resource (KARE) (*Cho et al., 2009*). To increase the validity of the study, we used the significant SNPs identified in previous BMI-related GWA-studies. Furthermore, we conducted SNP selection based on 10-fold cross validation used in conjunction with the wGRS on a SNP selection set consisting of 5,410 samples from Korean subjects. The wGRS of the selected SNPs was applied to an independent validation set consisting of 601 samples.

## METHOD

### Study subjects

The genotype data used in the present study were obtained from the KARE project (*Cho et al., 2009*). This study approved by Public Institutional Bioethics Committee designated by the Ministry of Health and Welfare (P01-201502-31-002). To ensure data quality, we eliminated samples and SNPs with a call rate of less than 98%. SNPs with a MAF of less than 0.05 were also excluded from our data set. A total of 6,011 samples (2,903 male and 3,108 female) were used for statistical analyses. The 6,011 samples were divided into one set of 5,410 samples (2,613 male and 2,797 female) to be used as a SNP selection set for 10-fold cross-validation and one set of 601 samples (290 male and 311 female) to be used as a validation set. The G*Power Version 3.1 software (Universität Kiel, Germany) (*Faul et al., 2009*) was used to calculate statistical powers. The software found both the test set ($n = 541$) and validation set ($n = 601$) to be at over 95%. Details on the number of samples are summarized in Table S1.

### SNP pruning for statistical analyses

In order to identify reliable SNPs for BMI prediction, we first identified the significant SNPs reported in prior BMI-related GWAS which had been validated by at least one

secondary replication study. Then, we obtained the genotype data for the collected GWAS markers including other markers from nearby regions ($\pm 10$ kb from the GWAS markers) from the KARE data (10,568 SNPs). To avoid the issue of high linkage disequilibrium (LD) found in the wGRS method, the LD coefficients ($r^2 > 0.2$) of all pairs of SNPs were calculated using the Haploview software (*Barrett et al., 2005*). Finally, we obtained a set of 193 SNPs which showed significant relationships with BMI in the previous GWA-studies. We also identified SNPs in the regions near the reported SNPs. The *P*-values obtained from regression analyses conducted on the training set ($n = 4,869$) were used to identify the most significant SNPs. The regression analysis was conducted using the GoldenHelix SVS8 software (Bozeman, MT, USA).

### SNP selection for BMI prediction

From the SNP selection set (5,410 samples), 10-fold cross validation was conducted on the genotype data (training set of 4,869 subjects and test set of 541 subjects) to identify the SNPs to be used for BMI prediction. The natural log-transformed BMI values were used for statistical analyses. SNPs were selected as tagging SNPs for each training set only where their *P*-value was less than 0.05 and where they were identified to be significant SNPs with the same LD. The wGRS was calculated as the sum of the number of BMI-increasing alleles multiplied by the regression slope across all variants in each set as previously described ($\sum_{i=1}^{n} Number\ of\ risk\ allele\ in\ SNP\text{i} \times Weight\text{i}$; $n =$ number of SNP, Weight: regression slope value of $SNP_i$) (*Hung et al., 2015*). Then, we divided the wGRS of each set into five sections in a subject number-dependent manner and calculated the average BMI values of the sections to obtain the trend lines. After 10-fold cross validation, we selected nine SNPs which overlapped across all training sets (Table S2). Detailed information and analysis of the selected SNPs are detailed in Table 1 and Fig. 1, respectively. The application process of wGRS using the selected nine SNPs to the independent validation set was the same as described above. The overall *P*-values for the trend lines were calculated using GraphPad Software (La Jolla, CA, USA) for all BMI values of each wGRS section.

## RESULTS

The average age and average BMI values were slightly higher among female subjects than male subjects across the complete set of samples, the SNP selection set, and the validation set. Detailed information of the subjects is given in Fig. 1 and Table S1.

An analysis flow chart for the present study is displayed in Fig. 1. The results of each set of 10-fold cross-validations show that the BMI values increased in all training sets and corresponding test sets with respect to wGRS for *P*-values of <0.0005 and $R^2$ of >0.2. The detailed *P*-values of the SNPs and BMI trend lines for the training set and test set are detailed in Table S2 and Fig. S1, respectively.

Of the 28 SNPs applied to the cross-validation process, we selected only nine SNPs which overlapped across all 10-fold cross-validations conducted for BMI prediction. The nine SNPs are listed in Table 1 with their location, allele information, and genotype data as obtained from the SNP selection set. We calculated the wGRS using the nine SNPs based on the SNP selection set ($n = 5,410$) and applied the wGRS to final validation set ($n = 601$).

**Table 1  Results of regression analysis and allele information using SNP selection set including additionally constructed sets.**

| Markers | Gene | Location | Position | P-values | | Allele information | | | | Genotype Count with BMI average | | | LD (r2>0.8) | GWAS catalog |
|---|---|---|---|---|---|---|---|---|---|---|---|---|---|---|
| | | | | Present study | GWAS catalog | BMI increasing | Minor | Major | MAF | C/C (BMI log.) | C/R (BMI log.) | R/R (BMI log.) | | |
| rs12566985 | FPGT-TNNI3K | 1:74536509 | Intron | $0.001 - 0.03$ | $2.00 \times 10^{-10}$ | G | A | G | 0.11 | 4,260 (3.197) | 1,061 (3.187) | 89 (3.169) | – | Felix et al. (2016) |
| rs6545809 | ADCY3 | 2:24903846 | Intron | $0.002 - 0.02$ | $6.00 \times 10^{-9}$ | T | T | C | 0.44 | 1,705 (3.191) | 2,660 (3.193) | 1,045 (3.204) | rs10182181 | Locke et al. (2015) |
| rs2943634 | – | 2:226203364 | Intergenic | $0.0002 - 0.02$ | $2.00 \times 10^{-14}$ | C | A | C | 0.08 | 4,577 (3.197) | 805 (3.182) | 28 (3.184) | – | Manning et al. (2012) |
| rs734597 | – | 6:50868566 | Intergenic | $0.0007 - 0.04$ | $3.00 \times 10^{-20}$ | A | A | G | 0.19 | 3,547 (3.192) | 1,680 (3.199) | 183 (3.202) | rs987237 | Speliotes et al. (2010) |
| rs11030104 | BDNF | 11:27662970 | Intron | $0.0004 - 0.006$ | $5.00 \times 10^{-19}$ | A | G | A | 0.45 | 1,617 (3.200) | 2,708 (3.195) | 1,085 (3.184) | – | Locke et al. (2015) |
| rs7988412 | GTF3A | 13:27426145 | Intron | $0.001 - 0.02$ | $2.00 \times 10^{-7}$ | T | T | C | 0.13 | 4,053 (3.192) | 1,275 (3.203) | 82 (3.197) | rs12016871 | Locke et al. (2015) |
| rs2241423 | MAP2K5 | 15:67794500 | Intron | $0.008 - 0.04$ | $1.00 \times 10^{-18}$ | G | G | A | 0.37 | 2,162 (3.192) | 2,508 (3.193) | 740 (3.205) | – | Speliotes et al. (2010) |
| rs7202116 | FTO | 16:53787703 | Intron | $0.00004 - 0.0006$ | $2.00 \times 10^{-10}$ | G | G | A | 0.13 | 4,143 (3.191) | 1,162 (3.206) | 105 (3.202) | – | Yang et al. (2012) |
| rs6567160 | – | 18:60161902 | Intergenic | $0.00006 - 0.002$ | $5.00 \times 10^{-30}$ | C | C | T | 0.24 | 3,101 (3.190) | 1,979 (3.198) | 330 (3.212) | – | Locke et al. (2015) |

**Notes.**

Gene name and location and position of the SNPs were obtained from the NCBI database. The P-values were calculated using logistic regression on the SNP selection set as well as on the additional reconstructed sets (10 sets). Hyphens(–) Indicate that there was no data available or it was not applicable. C/C, C/R, and R/R represent the homozygote of the major allele, and the heterozygote and homozygote of the minor allele, respectively.

BMI, Body Mass Index; MAF, Minor allele frequency; LD, Linkage Disequilibrium.

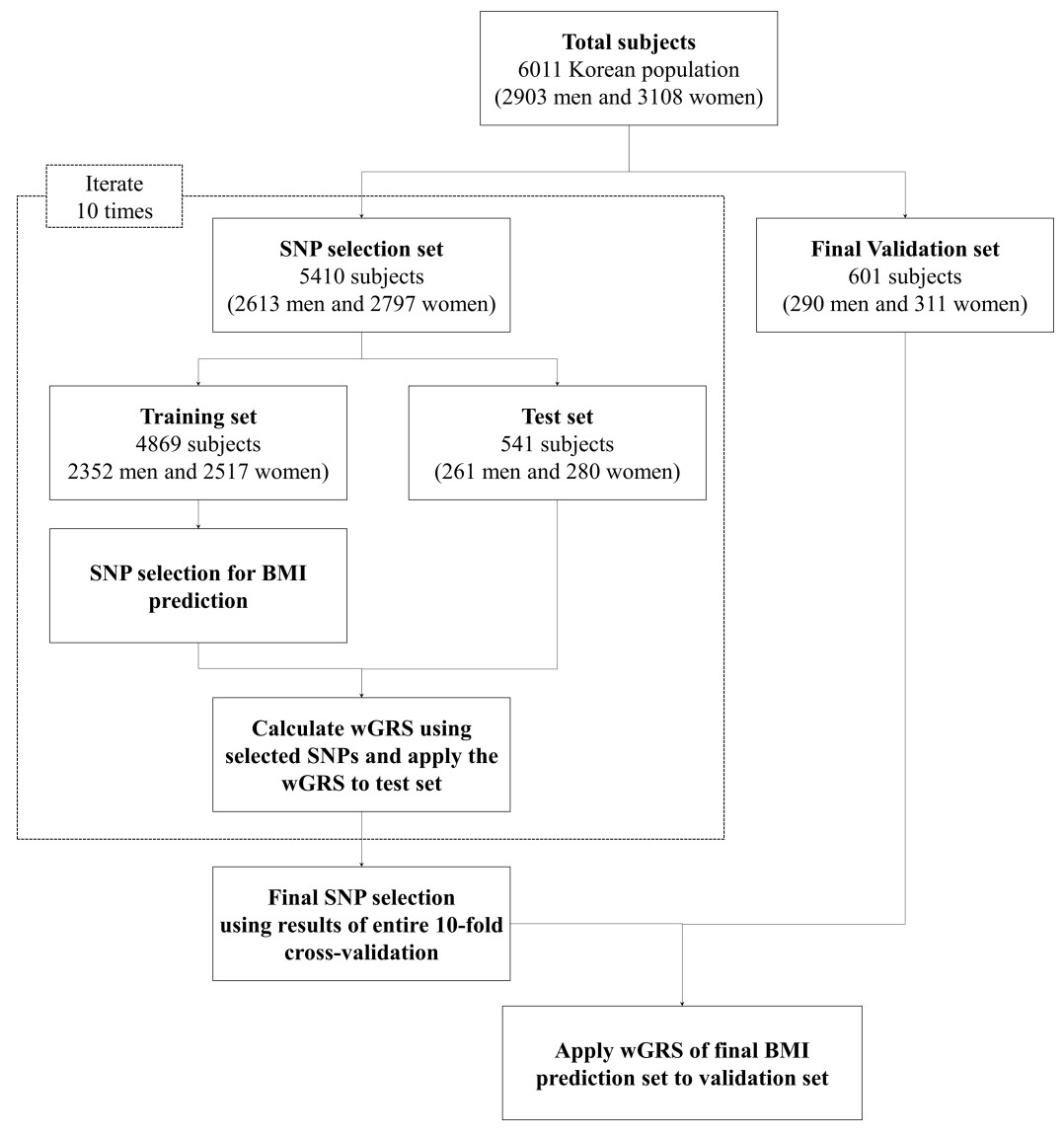

**Figure 1** Analysis flow chart used in the present study.

As expected, the results of the SNP selection set showed upward BMI trends across all three groups with $P$-values of 0.002 for the complete set, 0.01 for males, and 0.008 for females, respectively (data not shown).

To confirm the upward BMI trends and significance of SNPs, we randomly re-constructed an additional nine sets which had the same SNP selection set size ($n = 5,410$) and final validation set size ($n = 601$). The results showed that the nine SNPs were significantly related to BMI across all additionally created sets ($P < 0.05$). The detailed range of $P$-values is listed in Table 1. The standard curves for the SNP selection sets also showed significant association with BMI ($P < 0.0001$) with $R^2$ values of 0.9729 for the complete set, 0.9296 for males, and 0.9598 for females (Fig. 2).

**A.**

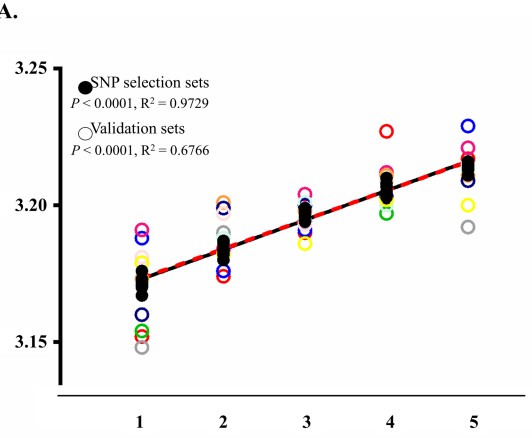

**B.**

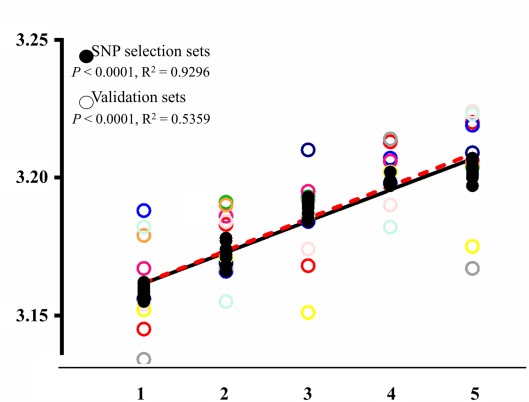

**C.**

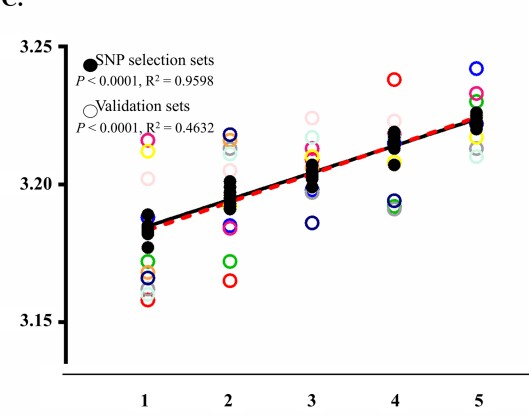

**Figure 2** **Analysis results applying the wGRS of the nine selected SNPs to the SNP selection set and to the final validation set.** The BMI trends of each analysis group are displayed using their standard curve with the overall $P$-values and $R^2$. The black closed and punctured circles represent the BMI values of each wGRS section in the SNP selection and validation set, respectively. The black and red dashed lines are the standard curves for the SNP selection and validation set. (A) BMI trends of SNP selection set and validation set using total subjects. (B) BMI trends of SNP selection set and validation set using male samples. (C) BMI trends of SNP selection set and validation set using female samples.

Application results of the wGRS to the final validation sets showed increasing trends with a $P$-value of <0.0001 and $R^2$ value of 0.6766 (Fig. 2A). This trend was also observed in the analyses for male ($P$-value of <0.0001 with $R^2$ of 0.5359) and female populations ($P$-value of <0.0001 with $R^2$ of 0.4632) (Figs. 2B and 2C).

## DISCUSSION

After the first large-scale GWAS was conducted using the KARE data with respect to several clinical traits (*Cho et al., 2009*), numerous follow-up studies have also been performed on independent cohorts. Of these studies, one reported that one SNP (*rs17178527* in *LOC729076*) was found to be significantly associated with BMI after applying Bonferroni correction with a $P$-value of $1.45 \times 10^{-7}$ (*Bae et al., 2016*). Several SNPs across various genes including *FTO*, *BDNF*, and *MC4R* were also found to be associated with BMI. Additional replication studies have consistently confirmed the significance of these SNPs in genes (*Cha et al., 2011*; *Cha et al., 2008*; *Hong et al., 2012*).

In the present study, we aimed to identify BMI-related SNPs using the KARE data in order to build a SNP set for BMI prediction. It is widely accepted that the influence of SNPs on BMI is small. Therefore, our study focused on increase of validity of using SNPs to understand BMI. We applied the wGRS to various test sets including validation sets and consistently found an increase in BMI trends. The $R^2$-values of the standard curves in the final validations (>0.4) suggest that the nine selected SNPs were not perfect indicators of BMI, but were significant enough to be considered for BMI prediction within the Korean population. Our results also indicated that the selected markers were not only associated to BMI within other populations but also within the Korean population.

There have been only a few studies conducted on BMI and/or obesity prediction in adulthood. We sought to find whether the 9 SNPs used in our models had been included in previous studies. From our research, we found that *rs2241423* in *MAK2K5* had been selected as one of SNPs for an obesity model for Caucasian populations (*Hung et al., 2015*). In addition, *rs11030104* in *BDNF* had been used in one study for BMI prediction within Korean populations (*Bae et al., 2016*). Two other *BDNF* SNPs (*rs6265* and *rs4923461*) had been used in prediction models and had been found to be in high LD ($r^2 > 0.8$) with *rs11030104*. Similarly, SNPs in *FTO* (*rs9939609* or *rs3751812*, $r^2 > 0.8$) and *ADCY3* (*rs6545814* or *rs10182181*, $r^2 > 0.8$) were used in other models (*Bae et al., 2016*; *Hung et al., 2015*; *Sandholt et al., 2010*). These results serve as evidence to support the validity of our results. In addition, comparison of two studies conducted on Korean populations suggest that at least 3 SNPs (*rs11030104* in *BDNF*, *rs6545814* in *ADCY3*, and *rs9939609* in *FTO*) might play crucial roles in BMI prediction for Korean populations.

There have been numerous reports which showing significant associations between BMI and SNPs in *BDNF*, *ADCY3*, and *FTO*. The functional role of several SNPs of various genes have also been revealed. Of the three SNPs *rs11030104*, *rs6545814*, and *rs9939609* which had been selected in both previous and the present study using Korean populations, *rs11030104* in *BDNF* was found to influence eating behavior, causing lower satiety responsiveness in children (*Monnereau et al., 2017*). In addition, a previous meta-analysis reported that

*rs6545814* in *ADCY3* was associated with BMI in East Asian populations (*Wen et al., 2012*). Although the functional role of *rs6545814* in BMI has not yet been fully demonstrated, one bioinformatics study reported that *rs6545814* was an expression quantitative trait loci of the *ADCY3* gene (*Yang et al., 2010*). Further, it was demonstrated that *rs9939609* in *FTO* had an impact on weight stabilization (*Woehning et al., 2013*).

One limitation in this study is that there is no further confirmation from other independent subjects. To rectify this limitation, we set aside 601 samples from the total 6,011 samples as a final validation set. Because of this limitation, this study could not identify specific markers for the Korean population, a step which would require additional validation sets. Future studies to build a more appropriate BMI prediction model using Korean-specific markers should be considered.

In summary, we identified nine BMI-related SNPs in a Korean population using the KARE data. Our results showed upward BMI trends across the samples using a 10-fold cross validation process. Application of our BMI prediction set to a final validation set showed a similar increasing BMI trend when using the wGRS. Although our study has some limitation as described above, the results from the present study might be useful for further BMI-related research.

### Funding
This work was supported by the Basic Science Research Program through the National Research Foundation of Korea funded by the Ministry of Education, Science and Technology (NRF-2015R1A2A1A15053987). This study was provided with biospecimens and data from the Korean Genome Analysis Project (4845-301), the Korean Genome and Epidemiology Study (4851-302), and Korea Biobank Project (4851-307, KBP-2015-035), which are supported by the Korea Center for Disease Control and Prevention, Republic of Korea. The funders had no role in study design, data collection and analysis, decision to publish, or preparation of the manuscript.

### Grant Disclosures
The following grant information was disclosed by the authors:
Ministry of Education, Science and Technology: NRF-2015R1A2A1A15053987.
Korean Genome Analysis Project: 4845-301.
Korean Genome and Epidemiology Study: 4851-302.
Korea Biobank Project: 4851-307, KBP-2015-035.

### Competing Interests
Hyun Sub Cheong is an employee of SNP Genetics, Inc. The authors declare there are no competing interests.

## Author Contributions

- Jin Sol Lee conceived and designed the experiments, analyzed the data, contributed reagents/materials/analysis tools, wrote the paper, prepared figures and/or tables, reviewed drafts of the paper.
- Hyun Sub Cheong conceived and designed the experiments, contributed reagents/materials/analysis tools, reviewed drafts of the paper.
- Hyoung-Doo Shin reviewed drafts of the paper.

## Ethics

The following information was supplied relating to ethical approvals (i.e., approving body and any reference numbers):

This study was provided with biospecimens and data from the Korean Genome Analysis Project (4845-301), the Korean Genome and Epidemiology Study (4851-302), and Korea Biobank Project (4851-307, KBP-2015-035), which are supported by the Korea Center for Disease Control and Prevention, Republic of Korea.

## Data Availability

The raw data (training set) has been uploaded as a Supplementary File.

## Supplemental Information

Supplemental information for this article can be found online at http://dx.doi.org/10.7717/peerj.3510#supplemental-information.

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
