# Peer review of "BMI prediction within a Korean population"

_PeerJ, doi:10.7717/peerj.3510_

## Round 0.1 · original submission · Major Revisions

Please carefully address the inspired reviewer's comments. I am especially concerned by the high proportion of R2 explained by SNPs.

Reviewer 1 ·

Basic reporting

Language needs extensive editing by a mother tongue professional. There are several gramar mistakes and sentences that need to be rephrased (auxiliar verbs are often lacking, for example; there are odd expressions such as "nine tenth" to say 90%; some sentences include gross errors, like in lines 70-71 of the manuscript and so on).
A huge part of the results session (105-109 and 112-119) and of the discussion session (139-158) contain methods information, indeed.

Experimental design

In the abstract it is necessary to clarify which "genotype data" the authors are speaking of.
It is not clear why authors tested several wGRSs across the ten training-test sets and finally built a unique wGRS with the 9 overlapping SNPs, instead of building the GRS directly with the single SNPs overlapping all the ten training-test sets.
However, the main question is another: authors present striking R2 values for the prediction of the BMI quintile by the GRS, in both men and women, while it is well-known that SNPs explain a small percentage of the BMI variance. Should this result be true, it would be revolutionary and should be the only point discussed in the manuscript, as it would mean that in the Korean adult population nine SNPs almost completely explain the BMI variance. How authors explain this? It is not clear which statistical test they used to obtain the above-mentioned R2 and the strong trends represented in Figure 2.

Validity of the findings

Authors use the discussion mainly to repeat methods and results but they do not comment the potential implications of their results adequately.

Additional comments

No comment.

Reviewer 2 ·

Basic reporting

Need for a native English speaking colleague review

Experimental design

Methods lack the description of the procedure to construct the wGRS.

Please explain why the authors have chosen 90% for selection and 10% for validation and state the power of the 2 sets.

Validity of the findings

I suggest to better underline the impact of this study.

Additional comments

IIn this paper Lee and coworkers have created a genetic risk score able to predict BMI in a Korean population.
The aim is sound, however there are several weakness mainly in statistical analyses as pointed above.

---

## Round 0.2 · accepted · Accept

The authors properly addressed the reviewer's comments.

Reviewer 1 ·

Basic reporting

No comment.

Experimental design

No comment.

Validity of the findings

No comment.

Additional comments

No comment.